# Development and Evaluation of a Reynolds-Averaged Navier–Stokes Solver in WindNinja for Operational Wildland Fire Applications

**Natalie S. Wagenbrenner \*, Jason M. Forthofer, Wesley G. Page and Bret W. Butler**

US Forest Service, Rocky Mountain Research Station, Missoula Fire Sciences Laboratory, 5775 W Highway 10, Missoula, MT 59808, USA; jason.forthofer@usda.gov (J.M.F.); wesley.g.page@usda.gov (W.G.P.); bret.butler@usda.gov (B.W.B.)

**\*** Correspondence: natalie.s.wagenbrenner@usda.gov

**Abstract:** An open source computational fluid dynamics (CFD) solver has been incorporated into the WindNinja modeling framework. WindNinja is widely used by wildland fire managers, as well as researchers and practitioners in other fields, such as wind energy, wind erosion, and search and rescue. Here, we describe the CFD solver and evaluate its performance against the WindNinja conservation of mass (COM) solver, and previously published large-eddy simulations (LES), for three field campaigns with varying terrain complexity: Askervein Hill, Bolund Hill, and Big Southern Butte. We also compare the effects of two model settings in the CFD solver, namely the discretization scheme used for the advection term of the momentum equation and the turbulence model, and provide guidance on model sensitivity to these settings. Additionally, we investigate the computational mesh and difficulties regarding terrain representation. Two important findings from this work are: (1) CFD solver predictions are significantly better than COM solver predictions at windward and lee side observation locations, but no difference was found in predicted speed-up at ridgetop locations between the two solvers, and (2) the choice of discretization scheme for advection has a significantly larger effect on the simulated winds than the choice of turbulence model.

**Keywords:** microscale wind modeling; RANS modeling; complex terrain; wildland fire

---

## 1. Introduction

WindNinja is a microscale diagnostic wind model, developed for and widely used in operational wildland fire applications both in the United States (U.S.) and abroad [1,2]. Microscale wind modeling is used for a variety of tasks in wildland fire management, including planning, reconstructing past events, and exploring what-if scenarios. Often, many, even thousands, of simulations, must be run in a short time frame, depending on the modeling objectives. WindNinja was developed over 15 years ago specifically for these types of tasks, and, to our knowledge, is the most widely used microscale wind model in wildland fire. WindNinja is embedded within a number of operational systems routinely used by U.S. Interagency Wildland Fire response teams, including the Wildland Fire Decision Support System [3] and FlamMap [4], and is also regularly used as a stand-alone model by both fire managers and on-the-ground firefighters.

The original version of WindNinja employs a numerical solver that enforces conservation of mass (hereafter referred to as the COM solver) to simulate mechanical effects of the terrain on the near-surface wind [1]. Evaluations against field data have shown that the COM solver can simulate many terrain-induced near-surface flow effects, including speed-up over ridges, terrain channeling, and reduced lee side velocities [1,2,5]. However, it is well-documented that COM solvers, including

the one in WindNinja, have difficulties simulating the flow field in regions where momentum effects dominate, notably on the lee side of terrain obstacles, where flow separation can lead to areas of recirculation [1,6].

Due to its success in the operational wildland fire community, WindNinja has been under continuous development over the last ten years, and has evolved into a wind modeling framework. This framework includes a graphical user interface, flexible initialization options, the ability to download data required for model initialization, user-selectable thermal parameterizations, and multiple output products. As a part of ongoing development efforts, a second numerical solver, based on computational fluid dynamics (CFD), has been added to the framework. This new solver is similar to the CFD model described by Forthofer et al. [1], but is based on free, open-source software embedded within the WindNinja framework. This new CFD solver is expected to improve predictions with only a marginal increase in computational effort, such that simulations are still affordable on typical laptop computers.

This paper describes the new CFD solver and provides an initial evaluation of its performance against field measurements, the COM solver in WindNinja, and previously published large-eddy simulation (LES) results. We investigate two commonly used discretization schemes for the advection term in the momentum equation, and three turbulence model configurations; the impact of these numerical settings on the results is assessed. The effect of the numerical mesh on results is also discussed. The specific goals of this study are to: (1) determine the most appropriate combination of numerical settings in the CFD solver for emergency response field applications and (2) compare the CFD solver predictions to predictions from the COM solver and LES observations.

## 2. WindNinja Framework

The WindNinja code is written primarily in the C/C++ programming language, and is open source and available on GitHub (github.com/firelab/windninja). It is cross-platform and runs on both the Linux and Windows operating systems. The framework includes a graphical user interface (GUI), command line interface (CLI), and an application programming interface (API). Additional model information can be found at weather.firelab.org/windninja.

WindNinja has seen broad and increasing usage (i.e., more than 7 million simulations in 30 countries during 2018). This is due, in part, to its simple user interface and suite of auxiliary features that ease model use. WindNinja has simple input requirements, which include a digital elevation model, vegetation type, and an initial wind. All of these inputs can be downloaded from online sources, via WindNinja. WindNinja allows three options for specification of the initial wind: (1) a domain-average wind (i.e., an average wind for the domain, specified at a single height above the ground); (2) a wind measured from one or more point observations (e.g., weather station data); and (3) a gridded wind field (e.g., winds from a coarser resolution model, such as a numerical weather prediction model).

The two numerical solvers are the core of the WindNinja framework. Both solve for a neutrally stratified flow; however, thermal parameterizations are available to approximate some thermal effects, including diurnal slope winds and non-neutral atmospheric stability. The slope flow parameterization is described in Forthofer et al. [7]. The stability parameterization adjusts the Gauss precision moduli in the governing equation solved in the COM solver. The Gauss precision moduli are adjusted, based on the estimated Pasquill stability class, following recommendations in Chan and Sugiyama [8] and Homicz [9]. As described in Forthofer et al. [1], the Gauss precision moduli control the relative amount of change allowed by the solver in the horizontal and vertical directions. The Gauss precision moduli are set to 1 if the stability parameterization is not used; this creates a numerical situation representative of neutral atmospheric conditions.

Since the current implementation of the stability parameterization is based on modifications to parameters in the governing equation solved in the COM solver, this parameterization is not available for use with the CFD solver. Future work intends to allow non-neutral simulations with the CFD

solver. The diurnal slope flow parameterization is incorporated into CFD simulations by first running a neutral CFD simulation, then adding the diurnal slope flow component to the CFD solution in each cell of the domain, and, finally, running a COM simulation on the slope flow-adjusted CFD solution. This chaining together of CFD and COM simulations allows the approximation of thermally driven slope flows without explicitly solving an energy equation in the CFD solver, which keeps the simulation times affordable.

## 3. CFD Solver Description

The CFD solver in WindNinja is based on OpenFOAM version 2.2.0 [10] (www.openfoam.org). The formulation of this solver is similar to that of the mass and momentum conserving solver, described in Forthofer et al. [1], which has been previously used in operational wildland fire applications under the name "WindWizard". Differences between the Fluent-based Forthofer et al. [1] solver and the CFD solver described here include the computational mesh structure, turbulence closure scheme, treatment of the ground boundary condition, and that all code used in the current CFD model is free and open source, which allows WindNinja to continue to be released without licensing restrictions or fees. This last point, regarding software licensing, is a major issue for operational wildland fire, particularly for government personnel who may not have access to funds or approval to purchase software licenses for their work.

As in Forthofer et al. [1], the flow is assumed to be steady, viscous, incompressible, turbulent, and neutrally-stratified, and the Coriolis force is ignored. WindNinja employs the simpleFoam solver, which is an implementation of the semi-implicit method for pressure-linked equations (SIMPLE) method, to approximate solutions to the steady-state, incompressible Reynolds-Averaged Navier–Stokes (RANS) equations. Using the Boussinesq approximation [11], the RANS equations are:

$$\frac{\partial \overline{u}_i}{\partial x_i} = 0 \tag{1}$$

$$\frac{\partial \left( \overline{u}_j \overline{u}_i \right)}{\partial x_j} = -\frac{1}{\rho} \frac{\partial \overline{p}}{\partial x_i} + \frac{\partial}{\partial x_j} \left( v \left[ \frac{\partial \overline{u}_i}{\partial x_j} + \frac{\partial \overline{u}_j}{\partial x_i} \right] \right) + \frac{\partial}{\partial x_j} \left( -\rho \overline{u_i' u_j'} \right) \tag{2}$$

In Equations (1) and (2), $\overline{u}_i$ and $\overline{u}_j$ are the time-averaged velocity components in the $i$ and $j$ coordinate directions, $u_i'$ and $u_j'$ are the instantaneous velocity components in the $i$ and $j$ coordinate directions, $p$ is pressure, $\rho$ is density, and $v$ is the laminar viscosity. A two-equation eddy viscosity turbulence model is used to model the contribution of the instantaneous velocity components. This introduces a turbulent viscosity, $v_t$, to account for the effects of the instantaneous velocity components:

$$\frac{\partial \left( \overline{u}_j \overline{u}_i \right)}{\partial x_j} = -\frac{1}{\rho} \frac{\partial \overline{p}}{\partial x_i} + \frac{\partial}{\partial x_j} \left( v + v_t \left[ \frac{\partial \overline{u}_i}{\partial x_j} + \frac{\partial \overline{u}_j}{\partial x_i} \right] \right) \tag{3}$$

Three two-equation turbulence models are investigated: the standard k-epsilon model [12], a modified k-epsilon model that allows the production and dissipation of turbulent kinetic energy (TKE) to be out of equilibrium at the ground, and the renormalization group (RNG) k-epsilon model [13]. In all cases, the turbulent viscosity is calculated as:

$$v_t = C_\mu \frac{k^2}{\varepsilon} \tag{4}$$

In Equation (4), $C_\mu$ is a constant (see Table 1), $k$ is the TKE, and $\varepsilon$ is the dissipation of TKE. Two additional transport equations are solved, one for $k$ and one for $\varepsilon$. For the standard k-epsilon model, the additional equations are:

$$\frac{\partial (k \overline{u}_i)}{\partial x_i} = \frac{\partial}{\partial x_j} \left[ \frac{v_t}{\sigma_k} \frac{\partial k}{\partial x_j} \right] + P - \varepsilon \tag{5}$$

$$\frac{\partial(\varepsilon\overline{u}_i)}{\partial x_i} = \frac{\partial}{\partial x_j}\left[\frac{v_t}{\sigma_\varepsilon}\frac{\partial\varepsilon}{\partial x_j}\right] + C_{\varepsilon1}\frac{P\varepsilon}{k} - C_{\varepsilon2}\frac{\varepsilon^2}{k} \qquad (6)$$

**Table 1.** Constants used in the governing equations.

| Parameter | Standard k-Epsilon | RNG k-Epsilon |
|:---:|:---:|:---:|
| $C_\mu$ | 0.09 | 0.085 |
| $\sigma_k$ | 1.0 | 0.7179 |
| $\sigma_\varepsilon$ | 1.3 | 0.7179 |
| $C_{\varepsilon1}$ | 1.44 | calculated |
| $C_{\varepsilon2}$ | 1.92 | 1.68 |
| $\beta$ | - | 0.012 |

In Equation (5), $P$ is the production of TKE and is given by:

$$P = 2v_t S_{ij} S_{ij} \qquad (7)$$

where $S_{ij}$ is the mean rate of strain tensor:

$$S_{ij} = \frac{1}{2}\left(\frac{\partial\overline{u}_i}{\partial x_j} + \frac{\partial\overline{u}_j}{\partial x_i}\right) \qquad (8)$$

The conservation equations are the same for the other two turbulence models, except the modified k-epsilon model uses a wall function for the production term in the dissipation equation and the RNG k-epsilon model treats the constant $C_{\varepsilon1}$ as a variable that depends on the ratio of the production of TKE to its dissipation:

$$C_{\varepsilon1RNG} = 1.42 - \frac{\eta(1 - (\eta/4.38))}{1 + \beta_{RNG}\eta^3} \qquad (9)$$

where:

$$\eta = \sqrt{P_k/\rho C_{\mu RNG}\varepsilon} \qquad (10)$$

and the production of TKE is:

$$P_k = \tau_{ij}\frac{\partial\overline{u}_i}{\partial x_j} = \mu_t\left(\frac{\partial\overline{u}_i}{\partial x_j} + \frac{\partial\overline{u}_j}{\partial x_i}\right)\frac{\partial\overline{u}_i}{\partial x_j} \qquad (11)$$

Model constants are listed in Table 1. The custom OpenFOAM code used in the modified k-epsilon model is available in the WindNinja GitHub repository.

The governing equations are discretized using the finite volume method. Two second-order discretization schemes for advection of the mean wind, linear upwind and the Quadratic Upstream Interpolation for Convective Kinematics (QUICK), are investigated in this work and described in Section 4.1. A first-order bounded Gauss upwind scheme is used for all other advection terms. A second-order Gauss linear limited discretization scheme is used for all diffusion terms.

The discretized equations are solved on a terrain-following, unstructured mesh with predominantly hexahedral cells (Figure 1). WindNinja employs a three-step meshing scheme using OpenFOAM mesh generation and manipulation utilities. The number of cells in the mesh is set based on a user-specified choice of mesh resolution. The four choices available to the user are 'coarse', 'medium', 'fine', and an option allowing the user to directly set the number of cells to use. The coarse, medium, and fine options correspond to 25 K, 50 K, and 100 K cells, respectively. In the first step of the meshing scheme, a blockMesh is generated above the terrain, using the blockMesh utility. Then, moveDynamicMesh is used to stretch the lower portion of the blockMesh down to the terrain. Finally, the near-ground cells are refined in all three directions using the refineMesh utility. The total number of cells are divided

equally between the blockMesh and the refined layer at the ground. The refineMesh utility is executed repeatedly until the specified number of cells have been allocated. This has proven to be a robust approach for automated meshing over complex terrain; however, there are limitations to this approach, which are discussed in Section 5.6. A comprehensive investigation of computational mesh quality is beyond the scope of this work, but key considerations regarding the current meshing algorithm are described for the reader and will be the focus of future work.

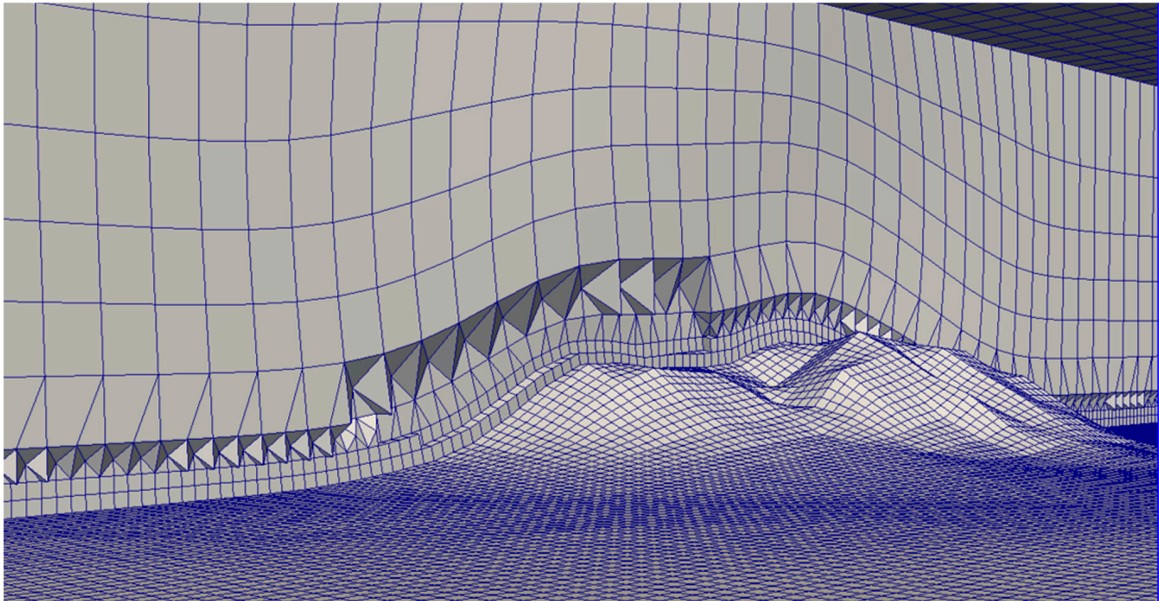

**Figure 1.** Slice through the computational mesh used for Big Southern Butte.

The inlet boundary conditions are specified as follows, per Richards and Norris [14]:

$$U = \frac{u_*}{\kappa_{k-\varepsilon}} ln\left(\frac{z}{z_0}\right) \tag{12}$$

$$k = \frac{u_*^2}{\sqrt{C_\mu}} \tag{13}$$

$$\varepsilon = \frac{u_*^3}{\kappa_{k-\varepsilon} z} \tag{14}$$

The friction velocity, $u_*$, is calculated as:

$$u_* = \frac{\kappa U_h}{ln\left(\frac{h}{z_o}\right)} \tag{15}$$

where $U_h$ is the input wind velocity at a specified height $h$ above the ground, and the von Karman constant, $\kappa$, is taken as 0.41.

The inlet is terrain-following. The non-inlet side boundaries are set to pressureInletOutlet for velocity and zero-gradient for TKE and dissipation of TKE. The pressureInletOutlet boundary condition assigns a zero-gradient condition if the flow is out of the domain and a velocity based on the flux in the cell face-normal direction if the flow is into the domain. The top boundary is specified as zero-gradient for velocity, TKE, and dissipation of TKE. A rough wall treatment is used for the ground boundary condition (nutkAtmRoughWallFunction). Wall functions are used to estimate shear stress at the ground based on the flow tangential to the ground in the near-ground cell. The wall function boundary conditions specified for TKE, dissipation of TKE, and velocity are zero-gradient (kqRWallFunction), a

turbulence dissipation wall constraint (epsilonWallFunction), and a Dirichlet condition (fixed value of 0), respectively. The roughness is set based on the vegetation selection in WindNinja, where the choices "grass", "brush", and "trees" correspond to a roughness of 0.01, 0.43, and 1.0 m, respectively.

We make two departures from the Richards and Norris [14] boundary condition recommendations: (1) we do not specify a shear stress at the top boundary and (2) we use a value of 0.41 for the von Karman constant, rather than the values determined by the turbulence model (which turn out to be 0.433 for the standard k-epsilon model, and 0.4 for the RNG k-epsilon model). These recommendations will be considered in future work.

The implemented boundary conditions were tested on a flat terrain case, and the inlet and outlet profiles are compared (Figure 2). The results shown in Figure 2 are for the standard k-epsilon turbulence model with the linear upwind discretization scheme. The horizontal extent of the computational mesh is $800 \times 400$ m with a top height of 80 m above sea level and cell horizontal spacing and cell height of 1 m in the near-ground cells. For a horizontally homogenous flat terrain, the inlet and outlet profiles should be identical. There is a slight decay in the velocity profile over the length of the domain (Figure 2), which could potentially be mitigated with specification of a shear stress, rather than zero-gradient, at the top boundary, as suggested by Richards and Norris [14]. The kink in the near-ground layer of the TKE profile is commonly observed in RANS modeling and may be due to one or more issues, including the near-ground cell height, inconsistency in the discretization used for TKE production term versus that used for the shear stresses in the momentum equation, or perhaps the turbulence model itself [14–16]. Future work will investigate improvements to the top boundary condition and approaches to mitigate the kink in the TKE profile, but, overall, these results are satisfactory for our typical use case in wildland fire applications.

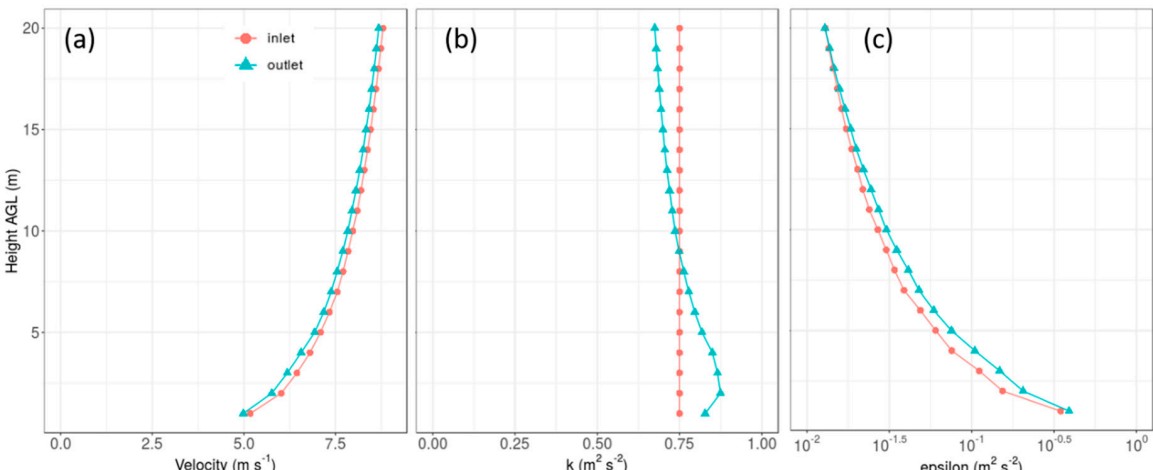

**Figure 2.** Profiles for (**a**) velocity, (**b**) turbulent kinetic energy (TKE), and (**c**) dissipation of TKE over flat terrain.

## 4. Methods

### 4.1. CFD Configuration and Settings Investigated

A mesh dependence study was carried out, comparing meshes with 100 K, 2 M, and 4 M cells (Appendix A). Results of the mesh dependence study show that the major features in the flow field are resolved with simulations from the mesh containing 100 K cells. The mesh dependence study also shows that the sensitivity of the results to the discretization scheme decreases as the mesh cell count increases. There are larger differences between discretization schemes in the coarser mesh. The purpose of the current work, however, is to evaluate the model as it would be used in an operational fire setting. Cell counts in the millions would require hours of simulation time on laptop computers used by field personnel, and, thus, would not be practical for emergency response applications like wildland fire.

Therefore, all CFD simulations were run with a fine mesh resolution, corresponding to 100 K cells. Mesh considerations and terrain representation are further discussed in Section 5.6. The diurnal slope flow parameterization was not used. The vegetation option was set to "grass", which corresponds to a roughness length of 0.01 m. The "domain average" initialization method was used to initialize the CFD simulations, using an average wind speed and direction measured at a single height above ground level at an upstream location at each site.

Two second-order discretization schemes are investigated for the advection of the mean wind, the linear upwind scheme and the QUICK scheme. The linear upwind scheme, which is the simplest and most commonly used second-order scheme, uses linear interpolation from the nearest upwind cell center [17]. The QUICK scheme uses a parabola to approximate the profile, using the two nearest upwind cell centers. Three k-epsilon-based turbulence models are investigated: the standard k-epsilon model, a modified k-epsilon model that allows production and dissipation of TKE to be out of equilibrium at the ground, and the RNG k-epsilon model, as described in Section 3. Table 2 summarizes the settings investigated and provides abbreviations for the six combinations used throughout the paper.

**Table 2.** Computational fluid dynamic (CFD) settings investigated.

| Abbreviation | Turbulence Model [1] | Discretization Scheme Used for Advection of Mean Wind [2] |
| --- | --- | --- |
| myKELU | modified k-epsilon | linear upwind |
| KELU | standard k-epsilon | linear upwind |
| RNGKELU | RNG k-epsilon | linear upwind |
| myKEQUICK | modified k-epsilon | QUICK |
| KEQUICK | standard k-epsilon | QUICK |
| RNGKEQUICK | RNG k-epsilon | QUICK |

[1] RNG is the renormalization group. [2] QUICK is the Quadratic Upstream Interpolation for Convective Kinematics.

### 4.2. COM Settings

WindNinja version 3.5.3 was used for the COM simulations. The diurnal slope flow parameterization was not used. The non-neutral stability parameterization was used only for the Askervein Hill case, which had slightly stable atmospheric conditions (see Section 4.3.1). As with the CFD solver, the fine mesh resolution option was used (which corresponds to 20K cells in the COM mesh). The vegetation option was set to "grass" and the "domain average" initialization method was used.

### 4.3. Field Observations

We evaluate the CFD and COM solvers against data from three field campaigns. Two are classic benchmark datasets, Askervein Hill [18,19] and Bolund Hill [20,21]. The third site, Big Southern Butte [22], represents a more complex geometry, with steeper slopes, higher ridgetops, and terrain bifurcations that are more representative of rugged terrain where wildland fires frequently occur. Big Southern Butte is surrounded by relatively flat terrain, however, which eases characterization of the approach flow and minimizes issues regarding model boundary conditions. Results are also compared with published LES results for Askervein Hill and Bolund Hill. We are not aware of published LES results for Big Southern Butte.

### 4.3.1. Askervein Hill

Askervein Hill (57°11.313′ N, 7°22.360′ W) is a geometrically simple hill rising 108 m above the surrounding terrain with a horizontal scale of about 3000 m (Figure 3a). Data were collected at 10 m above ground level along three transects, Lina A, Line AA, and Line B (Figure 3a). The MF03-D and TU03B datasets [19] are used for evaluations. The average approach flow measured at a reference location 3 km upstream was 8.9 m s$^{-1}$ from a direction of 210°. The atmospheric stability was slightly stable (Figure 3b) with average Richardson numbers between −0.0110 and −0.0074. The ground

roughness length was estimated as 0.03 m [23]. Elevation data at 23 m horizontal resolution on a 6 × 6 km domain from Walmsley and Taylor [24] are used for the simulations.

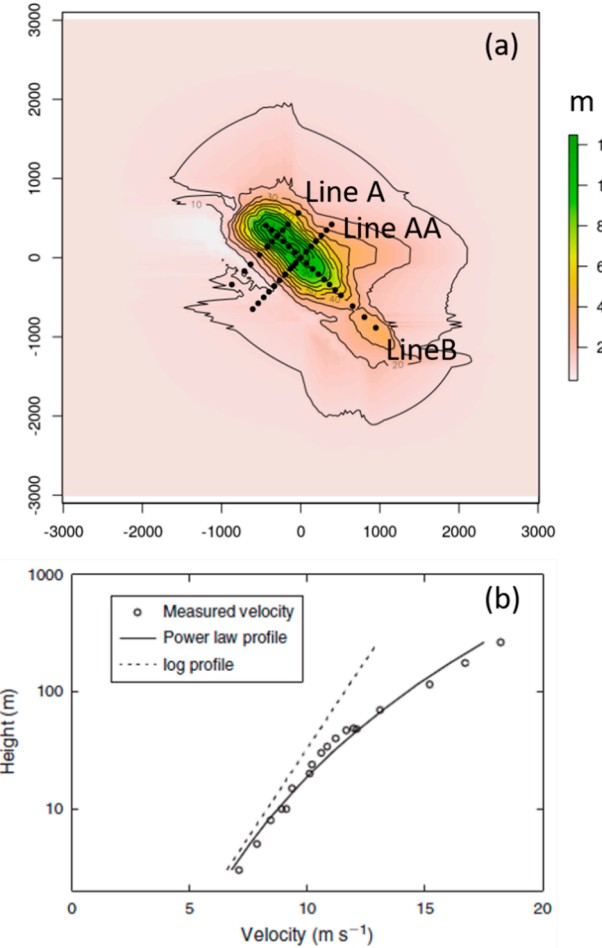

**Figure 3.** Askervein Hill (**a**) terrain and measurement locations with axes labeled in meters, with north toward the top of the figure and (**b**) the observed velocity profile measured at an upwind reference station compared to logarithmic and power law profiles; reproduced with permission from Forthofer et al. [1].

Characteristics of the computational mesh are shown in Table 3. The horizontal extent of the CFD computational mesh is 6 × 6 km, with the hill roughly centered in the domain. The mesh top height is 727 m above sea level (Table 3). The average horizontal spacing and cell height of the near-ground cells is 20 m. The COM mesh has the same horizontal extent as the CFD mesh, but has a 742 m top height, 43 m horizontal spacing, and a cell height of 0.4 m in the near-ground cells. The non-neutral stability parameterization was used for the COM simulation to approximate a slightly stable atmosphere, as measured at the upstream reference site.

**Table 3.** Computational mesh characteristics.

| Site | Solver [1] | Top Height ASL [2] (m) | Horizontal Grid Spacing (m) | Near-Ground Cell Height (m) |
|---|---|---|---|---|
| Askervein Hill | CFD | 727 | 20 | 20 |
| | COM | 742 | 43 | 0.4 |
| Bolund Hill | CFD | 92 | 3.8 | 3.8 |
| | COM | 26 | 4 | 0.1 |
| Big Southern Butte | CFD | 4318 | 68 | 68 |
| | COM | 2508 | 138 | 1.6 |

[1] CFD is computation fluid dynamics and COM is conservation of mass. [2] ASL is above sea level.

### 4.3.2. Bolund Hill

Bolund Hill (55°42.21′ N, 12°5.892′ E) is smaller than Askervein Hill, with only 12 m of relief and a horizontal scale of about 200 m, but it has a steep, cliff-like west face, which makes its geometry slightly more complex (Figure 4). Measurements were made along two transects, Line A and Line B (Figure 4). Three cases from the blind comparison study, described in Bechmann et al. [21], are chosen for this work (Table 4). The chosen cases are Cases 1, 3, and 4, which correspond to wind speeds and directions of 10.9 m s$^{-1}$ from 270°, 8.7 m s$^{-1}$ from 239°, and 7.6 m s$^{-1}$ from 90°, respectively. The upstream roughness was estimated as 0.0003 m for Cases 1 and 3 (approach flow over water) and 0.015 m for Case 4 (approach flow over land) [21]. Atmospheric stability was characterized as near-neutral for all three cases [21]. Elevation data with a horizontal resolution of 0.25 m and a horizontal extent of 800 × 400 m are used for the simulations.

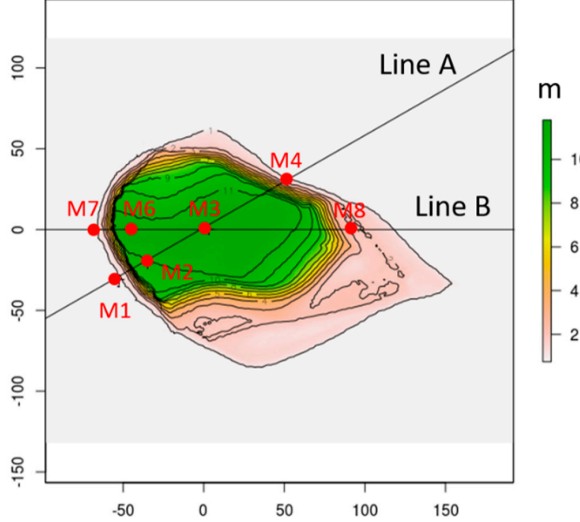

**Figure 4.** Bolund Hill terrain and measurement locations. Axes labels are in meters, and north is toward the top of the figure.

**Table 4.** Bolund Hill cases investigated.

| Case | Wind Speed (m s$^{-1}$) | Wind Direction (°) |
|------|-------------------------|--------------------|
| 1    | 10.9                    | 270                |
| 3    | 8.7                     | 239                |
| 4    | 7.6                     | 90                 |

The CFD mesh has a horizontal extent of 800 × 400 m, with the hill centered in the domain. The mesh top height is 92 m above sea level (Table 3). The average horizontal spacing and cell height of the near-ground cells is 3.8 m (Table 3). The COM mesh has the same horizontal extent as the CFD mesh, but has a top height of 26 m, 4 m horizontal grid spacing, and a near-ground cell height of 0.1 m (Table 3).

### 4.3.3. Big Southern Butte

Big Southern Butte (43°24.083′ N, 113°01.433′ W) is a tall, isolated mountain and substantially more geometrically complex than Askervein Hill or Bolund Hill (Figure 5). It has a vertical relief of 800 m and a horizontal scale of about 4 km. The butte is characterized by a mix of slope angles and multiple bifurcations, with ridges and valleys of various sizes forming the sides of the butte. As with Askervein and Bolund hills, the butte is covered predominantly by grass, although there are scattered trees in some locations at the higher elevations. The butte is surrounded by flat terrain covered by grass and small shrubs for more than 50 km in all directions.

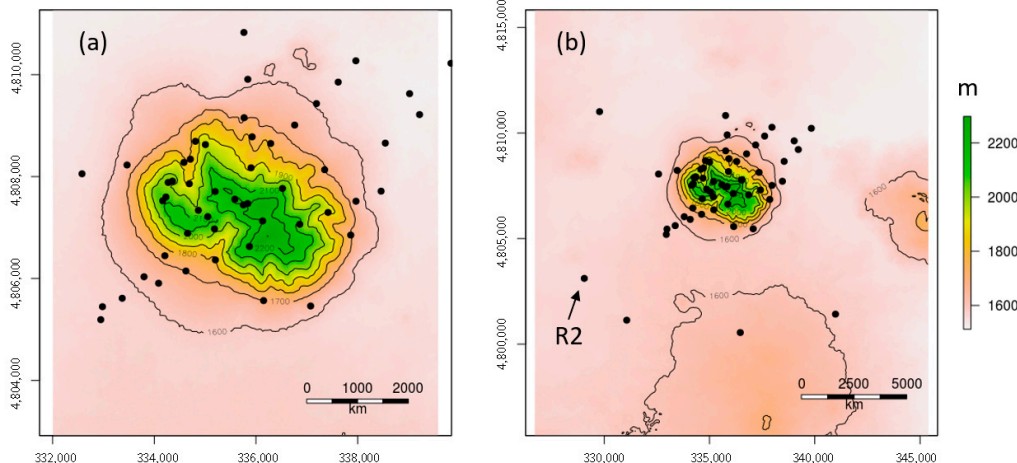

**Figure 5.** Big Southern Butte terrain and measurement locations. Panel (**a**) is zoomed in on the butte and (**b**) shows the full study area and the location of reference sensor R2. Axes labels are in meters and north is toward the top of the figure.

The data used for evaluation were collected during the field campaign described in Butler et al. [22]. Wind speed and direction were measured at 3 m above ground level, at 53 locations on and around the butte (Figure 5). Here, we use the 10 min averaged winds at 1700 LT on 18 July 2010 as the evaluation case. This is the same case as the externally forced flow event investigated by Wagenbrenner et al. [5]. During this period, the approach flow was relatively steady (Figure 6b,c) and wind speeds were moderately strong (Figure 6a,b), creating near-neutral atmospheric stability conditions at the surface. The average wind measured at the upstream reference station, R2 (Figure 5b), was 8.3 m s$^{-1}$ from 222° (Figure 6b,c). Elevation data from the Shuttle Radar Topography Mission (SRTM) dataset [25], covering an extent of 19 × 20 km at 30 m horizontal resolution, are used for the simulations.

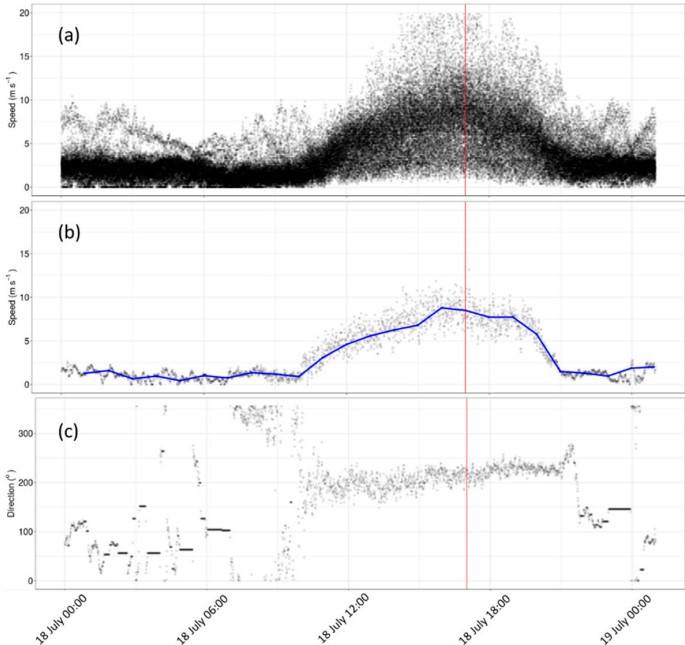

**Figure 6.** Instantaneous wind speeds measured at Big Southern Butte on 18 July 2010 at (**a**) all sensors and (**b**) sensor R2 and (**c**) instantaneous wind direction measured at sensor R2 on 18 July 2010. The blue line indicates 10 min averaged wind speed at the top of each hour. The red line indicates 1700 LT.

The CFD mesh has a horizontal extent of 19 × 20 km, with the butte centered in the domain. The mesh top height is 4318 m above sea level (Table 3). The average horizontal grid spacing and cell height of the near-ground cells is 68 m (Table 3). The COM mesh has the same horizontal extent as the CFD mesh but has a top height of 2508 m, 138 m horizontal grid spacing, and a near-ground cell height of 1.6 m (Table 3).

*4.4. Evaluation Methods*

One goal of this study is to determine the most appropriate combination of numerical settings for the CFD solver. Results from the six combinations of numerical settings used in the CFD solver are explored by inspecting raster outputs of the predicted surface wind speeds under each combination of numerical settings at each site. Observed and predicted winds along transects at each site are also inspected. Model performance for the CFD and COM solvers is quantified in terms of the root mean square error (RMSE), mean bias error (MBE) and mean absolute percent error (MAPE):

$$RMSE = \left[ \frac{1}{N} \sum_{i=1}^{N} \left( \varphi_i' \right)^2 \right]^{1/2} \tag{16}$$

$$MBE = \frac{1}{N} \sum_{i=1}^{N} \varphi_i' \tag{17}$$

$$MAPE = \frac{1}{N} \sum_{i=1}^{N} \frac{\left| \varphi_i' \right|}{\varphi_i} \times 100 \tag{18}$$

where $\phi$ is the observed value, $\phi'$ is the difference between predicted and observed, and $N$ is the number of observations. Results from LES conducted by others are included in transect plots for Askervein Hill and Bolund Hill for visual comparisons. The LES predictions are shown as a reference, but are not included in the statistical analyses.

Analyses at Askervein and Bolund hills focus on comparisons of observed and predicted wind speed, rather than wind direction. This is primarily because, with the exception of Case 4 at Bolund Hill, the observed data do not include major recirculation regions or other terrain-induced directional changes in the wind to warrant that analysis. The observed flow field at Big Southern Butte is much more complex, with multiple recirculation regions and flow channeling around the butte, as well as within side drainages on the butte [5,22]. Therefore, analysis at Big Southern Butte includes comparisons of wind speeds and directions along selected transects roughly parallel to the prevailing wind direction as well as with the full set of observations collected on and around the butte. Although wind direction data are presented for Big Southern Butte, mostly to provide additional context regarding the flow dynamics over the butte, the focus of this work is wind speed prediction. Future work will specifically explore simulated lee side flow dynamics and representation of flow separation and recirculation.

An Analysis of Variance (ANOVA) is used to determine the relative effect of the CFD settings on wind speed error. Specifically, the variability in the dependent variable (predicted–observed) is compared to the effects of three independent variables: the discretization scheme (two levels), turbulence model (three levels), location (three levels), and all two-way interactions at the three field sites. The three location levels correspond to either the windward, ridgetop, or leeward locations of the observations. Square-root and cube-root transformations are applied where necessary to meet the assumptions of normality and homoscedasticity of the residuals. The family-wise error rate for multiple comparisons between the means of the various factors levels is controlled using Tukey's Honest Significant Difference method [26]. The effect size of each individual independent variable is compared by using the Eta-squared ($\eta$2) statistic, as computed by the sjstats package in R [27], which is a measure of the proportion of the total variation in the dependent variable that can be attributed to a specific independent variable.

The data are also pooled across all three field sites to assess the relative effects of the discretization scheme, turbulence model, location, and solver type (i.e., COM vs. CFD) on predicted error. In this case, a linear mixed-effects model is constructed using the lmer function in the lme4 package in R [28]. The fixed effects are the discretization scheme, turbulence model, location, and solver type, while the random effect was the field site. The relative importance of the independent fixed-effect variables is assessed using the relaimpo package in R [29], which estimates the proportion of the variance explained by the model, due to the independent variables.

## 5. Results and Discussion

### 5.1. Askervein Hill

#### 5.1.1. CFD-Predicted Flow Patterns in the Horizontal Plane

The CFD-predicted 10-m wind speeds using each of the six combinations of numerical settings are shown in Figure 7. Several notable flow features are evident. All combinations predict a reduction in speed as the flow approaches the hill, speed-up on the ridgetop, and reduced speeds on the lee side of the hill. The size, magnitude, and shape of each of these regions in the predicted flow field vary with the choice of numerical setting. Noticeably, the choice of discretization scheme appears to have a bigger impact on the flow than the choice of turbulence model, both in terms of the magnitude of predicted speeds, and spatial patterns in the flow field, particularly on the lee side of the hill (Figure 7a–c versus d–f).

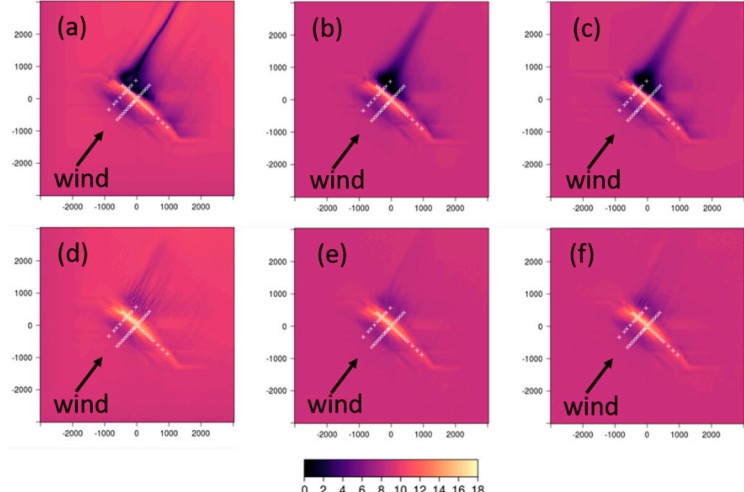

**Figure 7.** CFD-predicted wind speeds in m s$^{-1}$ at 10 m above ground level over Askervein Hill using (**a**) myKELU; (**b**) KELU; (**c**) RNGKELU; (**d**) myKEQUICK; (**e**) KEQUICK; (**f**) RNGKEQUICK. White crosses indicate measurement locations. Black arrows denote the prevailing wind direction. Axes labels are in meters.

The linear upwind scheme produces less ridgetop speed-up and more speed reduction in the lee of the hill, as compared with the QUICK scheme (Figure 7a–c versus d–f). The region of reduced speeds in the immediate lee of the hill is also part of a broader, more coherent pattern in the flow field in the linear upwind simulations, as compared with the same region in the QUICK simulations.

Low-velocity streamwise streaks are visible in the flow field on the lee side of the hill for all combinations of numerical settings. The linear upwind scheme produces a broad region of low-velocity flow behind the hill, with a streak extending far downwind of this region (Figure 7a–c). The QUICK scheme produces multiple, narrower streaks in the immediate lee of the hill, as compared with the linear upwind scheme (Figure 7d–f). The streaks are most well-defined (sharpest gradient normal to

the streak) in the myKE simulations (Figure 7a,d). The KE and RNGKE turbulence models appear to smear out the streaks, as compared with the myKE model (Figure 7b,c and e,f versus a and d).

There is experimental and observational evidence from both turbulence and geomorphological research to suggest that the predicted streamwise low-velocity streaks are real, terrain-induced features in the flow field [30–34]. Using RANS modeling, Hesp and Smyth [34] show that, for high Reynolds number flows, dune-shaped terrain features induce paired counter-rotating vortices within the wake region of the mean flow. The paired counter-rotating vortices are the mean flow manifestation of transient von Karman vortex shedding (i.e., alternating detachment of vortices on the lee side of a blunt isolated object). Hesp and Smyth [34] further show that the shape and aspect ratio of the terrain feature affects the structure of the horizontal and vertical flow within the wake region. The hills investigated in this work can be broadly categorized as dune-shaped, and, indeed, our simulations also contain paired counter-rotating vortices in the wake zone. The lee side streamwise streaks, visible in our simulations, are the convergence zones of these paired vortices.

We conclude that the streamwise streaks visible in our simulations are the result of simulated converging counter-rotating vortices within the wake regions; however, it is not clear how strong and well-defined the streaks should be. Development of the most well-defined streaks with the strongest cross-flow gradients (Figure 7a,d) could indicate insufficient turbulent diffusion in the model. If that is the case, then modeling choices which smear out the streaks to some degree would be desirable. Other CFD modeling studies have also reported streaks with varying patterns and strengths, associated with topographical features in RANS and time-averaged LES simulations (e.g., [35]), but there appears to be little guidance in terms of the realistic representation of these streamwise flow features.

### 5.1.2. Comparisons with Observations

Inspection of the speed-up profiles along the transects further indicates that the choice of discretization scheme has a bigger effect on the predictions than the choice of turbulence model, particularly on the lee side of the hill (Figure 8). This is indicated by the tight clustering of lines depicting simulations using the linear upwind scheme (red, orange, and pink lines) versus the QUICK scheme (blue, green, and light blue lines) (Figure 8). The LES results from Golaz et al. [36] generally compare better with observations than the CFD results do, particularly on the lee side. The LES results are similar to the COM results on the ridgetop locations, although LES over-predicts at the ridgetop in Line AA (Figure 8b).

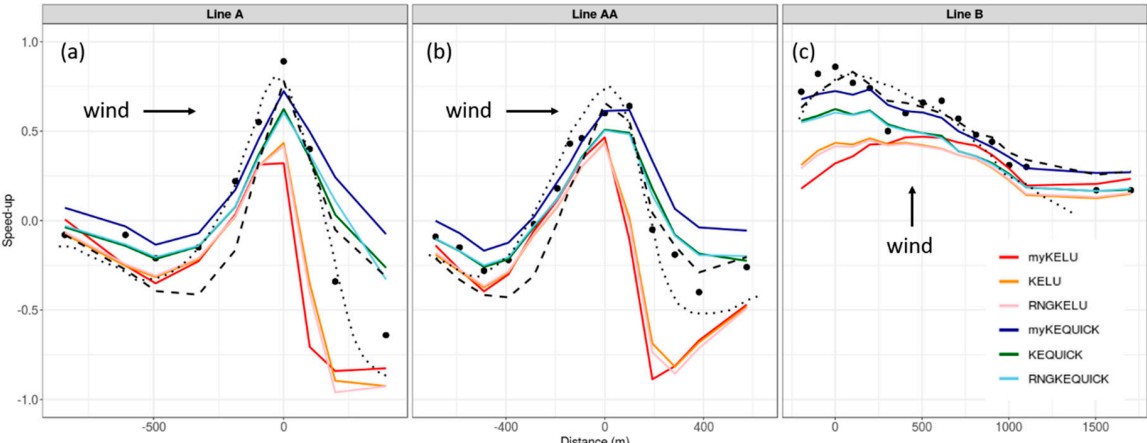

**Figure 8.** Model comparisons with observed data at Askervein Hill for (**a**) Line A; (**b**) Line AA; and (**c**) Line B. Black circles are observed data. Black dashed lines are COM solver results. Dotted black lines are large-eddy simulation (LES) results, redrawn from Golaz et al. [36]. The x-axis is distance along the transect. The y-axis is fractional speed-up, relative to the observed speed at a reference station upwind.

Compared to the linear upwind scheme, the QUICK scheme, on average, predicts higher speeds at the ridgetop (13.2 versus 11.8 m s$^{-1}$, $p$ = 0.0086) and leeward (9.15 versus 2.49 m s$^{-1}$, $p$ < 0.0001) locations, which is consistently in better agreement with observations (MAPE of 7–42% versus 15–64%, respectively) (Table 5). The QUICK scheme over-predicts on the lee side by 2.1 m s$^{-1}$, while the linear upwind scheme under-predicts by 4.5 m s$^{-1}$. The linear upwind scheme also under-predicts at the ridgetop and windward locations by 2.2 and 1.0 m s$^{-1}$, respectively. These results suggest that the QUICK scheme outperforms the linear upwind scheme at all locations; however, atmospheric stability was slightly stable during the observation period, so a model simulating neutral conditions, like the CFD solver used here, would be expected to under-predict, particularly at ridgetop locations.

**Table 5.** Model root mean square error (RMSE), mean bias error (MBE), and mean absolute percent error (MAPE) for wind speeds at windward (w), ridgetop (r), and leeward (l) sensor locations at Askervein Hill. Positive MBE indicates model over-prediction. Bold font indicates the lowest error at each location.

| Location | Settings [1] | RMSE | MBE | MAPE (%) |
|----------|--------------|------|-----|----------|
| w | LU | 1.23 | −1.04 | 21 |
|   | QUICK | **0.79** | **−0.19** | **6.1** |
|   | COM | 1.9 | −1.76 | 20 |
| r | LU | 2.80 | −2.22 | 15 |
|   | QUICK | 1.21 | −0.85 | 6.9 |
|   | COM | **0.69** | **0.06** | **4.4** |
| l | LU | 5.05 | −4.53 | 64 |
|   | QUICK | 2.64 | 2.13 | 42 |
|   | COM | **1.58** | **1.10** | **26** |

[1] LU is linear upwind; QUICK is the Quadratic Upstream Interpolation for Convective Kinematics, and COM is conservation of mass.

The COM solver with the non-neutral stability parameterization enabled predicts the ridgetop speeds well (MAPE of 4%), but over-predicts on the lee side of the hill, particularly for Line A (Figure 8a), resulting in a MAPE of 26%. The COM solver performs better, in terms of the MAPE, at both the ridgetop and leeward locations, than the linear upwind (15% and 64%, respectively) and QUICK (6.9% and 42%, respectively) simulations (Table 5).

The majority of the error in predicted wind speed in the CFD results is attributed to the discretization scheme and its interaction with location, rather than the choice of turbulence model. Specifically, 25% of the variation in wind speed error is due to the discretization scheme ($\eta2$ = 0.25), as opposed to the choice of turbulence model, which explained less the 1% of the variation ($\eta2$ < 0.01). The location of observation also had a significant effect on wind speed error with the largest errors across all settings occurring at the lee side locations, which accounted for about 12% ($\eta2$ = 0.12) of the total variation in wind speed error (Figure 8).

*5.2. Bolund Hill*

5.2.1. CFD-Predicted Flow Patterns in the Horizontal Plane

Similar flow features are visible in the CFD-predicted 5 m wind speeds (Figures 9–11) as those reported for Askervein Hill in Section 5.1.1. In all cases, and for all combinations of numerical settings, there is a reduction in speed as the flow approaches the hill, a ridgetop speed-up, and reduced speeds on the lee side of the hill. As in the Askervein Hill simulations, the size and magnitude of each of these flow regions varies with the choice of numerical setting, and the choice of discretization scheme appears to have a larger impact on the flow than the choice of turbulence model. The linear upwind scheme produces a broader, more coherent region of reduced speeds on the lee side of the hill than the QUICK scheme, which produces narrower, streamwise fingers of reduced speeds in the immediate lee

of the hill. The same low-velocity streamwise streaks are visible in the flow field on the lee side of the hill for all combinations of numerical settings, and, as with the Askervein Hill simulations, the myKE simulations have the strongest cross-streak gradient. This is most apparent in the simulations for Case 4, where the wind is coming from the east and the steep, cliff-like west face is the lee side of the hill (Figure 11).

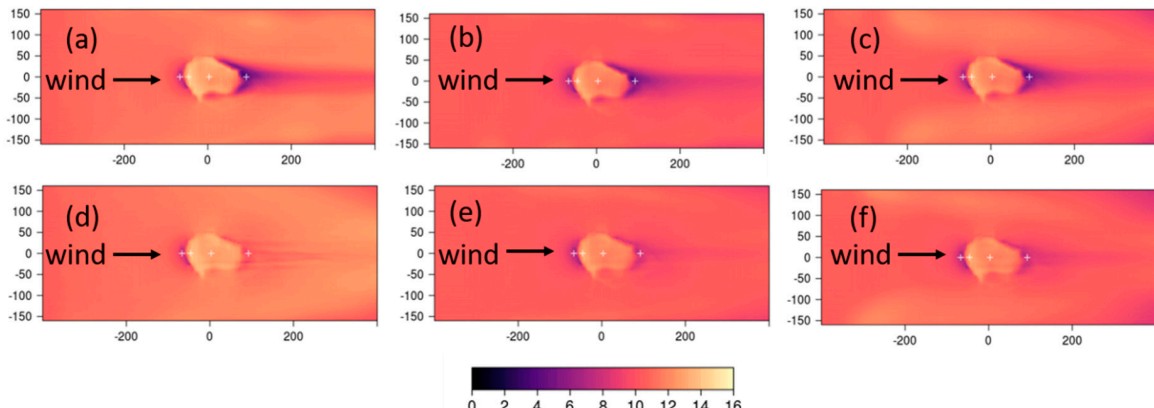

**Figure 9.** CFD-predicted wind speeds in m s$^{-1}$ at 5 m above ground levle over Bolund Hill for Case 1, using (**a**) myKELU; (**b**) KELU; (**c**) RNGKELU; (**d**) myKEQUICK; (**e**) KEQUICK; (**f**) RNGKEQUICK. White crosses indicate measurement locations. Black arrows denote the prevailing wind direction. Axes labels are in meters.

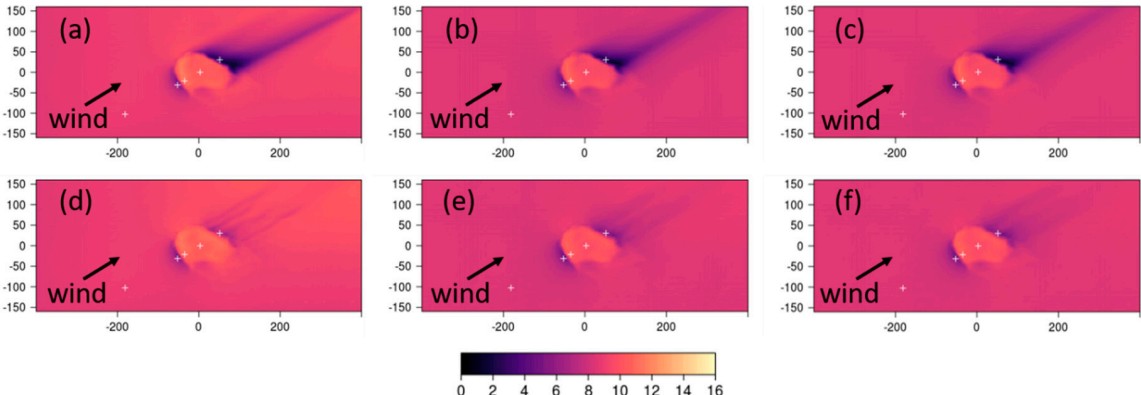

**Figure 10.** Same as Figure 9, but for Case 3.

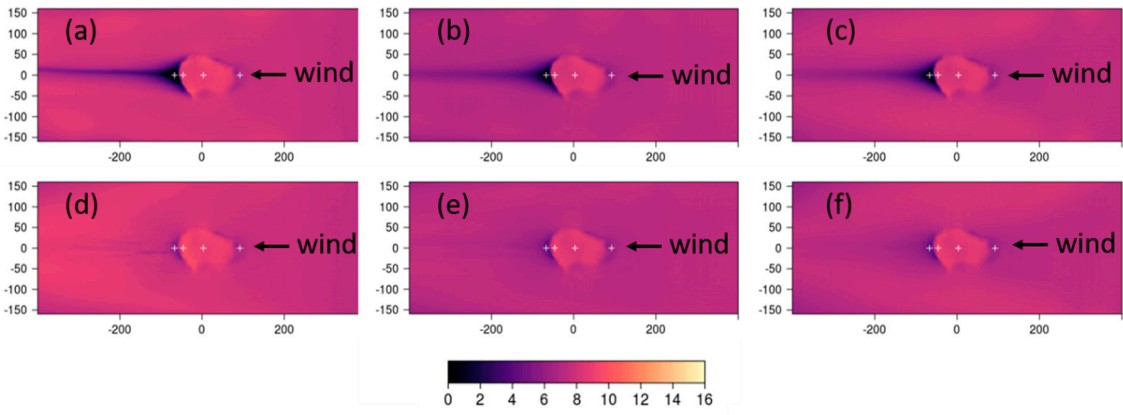

**Figure 11.** Same as Figure 9, but for Case 4.

### 5.2.2. Comparisons with Observations

Like the Askervein Hill results, inspection of the speed-up profiles for the Bolund Hill transects indicates that the choice of discretization scheme has a bigger effect on the predictions than the choice of turbulence model. This is indicated by the tight clustering of lines depicting simulations using the linear upwind scheme (red, orange, and pink lines), versus the QUICK scheme (blue, green, and light blue lines), especially in the lee of the hill (Figure 12).

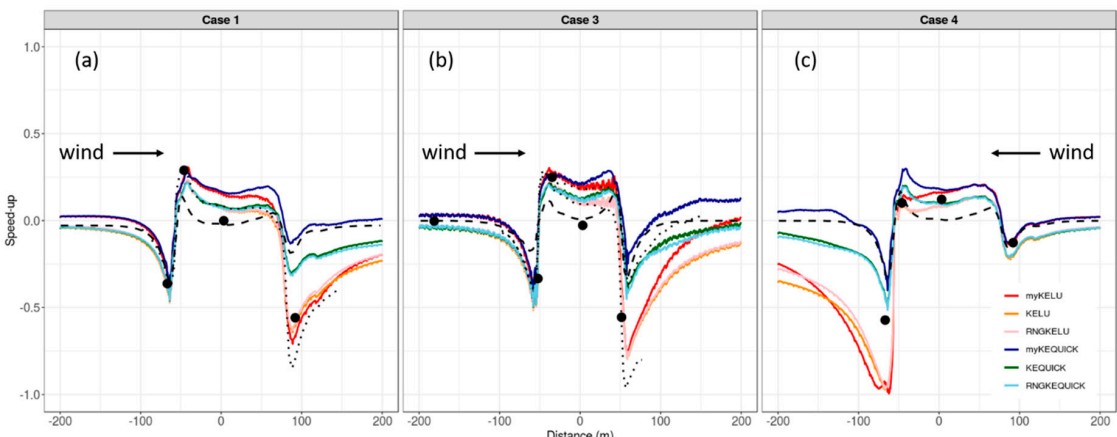

**Figure 12.** Model comparisons to observed data at Bolund Hill for (**a**) Case 1; (**b**) Case 3 and; (**c**) Case 4. Black circles are observed data. The x-axis is distance along the transect. The y-axis is fractional speed-up relative to the observed speed at a reference station upwind. Black dashed lines are the COM solver results. Dotted black lines are LES results redrawn from Bechmann et al. [21] and Vuorinen et al. [37].

For Case 1, all of the models predict the reduced speed in the approach flow and speed-up at the ridgetop (Figure 12a). The COM solver has the best prediction at the mid-location on the hill, with the LES, KE and RNGKE simulations slightly over-predicting at this location. The myKE simulations have the worst predictions at this mid-hill location, compared to the other models. In the lee of the hill, the COM simulation is the worst performer, and largely over-predicts the lee side speed. All the linear upwind predictions are similar in the lee of the hill, and slightly under-predict at this location. The LES simulation is similar to the linear upwind simulations at this lee side location, but had a slightly larger under-prediction.

The results are similar for Case 3, with all models comparing well at the first two observation locations along the mean wind direction (Figure 12b), and all, except the COM simulation, over-predicting at the mid-hill location. The COM solver does not produce enough reduction in speed in the approach flow, but predicts speed-up at the ridgetop and the reduction in speed at the mid-hill location well when compared to the observations. The COM simulations and the QUICK simulations all over-predict on the lee side. The lee side reduction in speed from the linear upwind simulations is closer to the observed reduction in speed. If anything, the linear upwind scheme simulations under-predict on the lee side. The LES simulations span the CFD simulations on the lee side of the hill, with one LES simulation over-predicting and the other under-predicting at this location.

Results for Case 4 are similar to those for Case 1 and 3, except that the under-predictions are larger on the lee side of the hill. This difference regarding the lee side in Case 4, compared to Cases 1 and 3, is likely due to the steep west face on the lee side of the hill. No published LES simulations were found for this case for comparison.

As opposed to the results from Askervein Hill, the evaluation metrics do not suggest that one particular set of CFD settings produce better wind speed predictions across all cases and locations (Table 6). However, consistent with the Askervein Hill results, the discretization scheme explains more variation in wind speed error than the choice of turbulence model ($\eta 2 = 0.07$ vs. $< 0.01$). The QUICK

scheme produces similar or lower MAPEs compared to the linear upwind scheme, except on the lee side of the hill, where the linear upwind scheme produces the lowest MAPE of 20% (Table 6). When averaged across all locations, the linear upwind scheme under-predicts wind speed by 0.75 m s$^{-1}$, while the QUICK scheme over-predicts by 0.21 m s$^{-1}$.

**Table 6.** Model root mean square error (RMSE), mean bias error (MBE), and mean absolute percent error (MAPE) for wind speeds at windward (w), ridgetop (r), and leeward (l) sensor locations at Bolund Hill. Positive MBE indicates model over-prediction. Bold font indicates the lowest error at each location.

| Location | Settings [1] | RMSE | MBE | MAPE |
|---|---|---|---|---|
| w | LU | 0.68 | −0.41 | 6.0 |
| | QUICK | **0.58** | **−0.27** | **5.2** |
| | COM | 1.08 | −0.39 | 6.9 |
| r | LU | 1.89 | −1.01 | 24 |
| | QUICK | **1.63** | −0.09 | **17** |
| | COM | 2.28 | **0.06** | 28 |
| l | LU | **1.09** | **−0.69** | **20** |
| | QUICK | 1.96 | 1.43 | 37 |
| | COM | 2.63 | 2.44 | 54 |

[1] LU is linear upwind; QUICK is the Quadratic Upstream Interpolation for Convective Kinematics, and COM is conservation of mass.

### 5.3. Big Southern Butte

#### 5.3.1. CFD-Predicted Flow Patterns in the Horizontal Plane

The differences between the linear upwind and QUICK discretization schemes are even more striking in the Big Southern Butte simulations than the Askervein Hill or Bolund Hill simulations (Figure 13). Consistent with the simulations at Askervein Hill and Bolund Hill, the linear upwind scheme produces a broader region of reduced speeds in the immediate lee of the butte with a narrow streak of low-velocity flow extending streamwise out of the domain. Narrow streamwise streaks of increased speed are also visible adjacent to the low-velocity streaks and extend out of the domain parallel to the low-velocity streaks.

As in the Askervein Hill and Bolund Hill simulations, the QUICK scheme produces narrow, well-defined streaks of low-velocity flow in the immediate lee of the butte (Figure 13d–f). In this case, the narrow streaks are noticeably wavier, especially for the myKEQUICK combination (Figure 13d), than those produced by the QUICK simulations at Askervein Hill and Bolund Hill. The QUICK scheme produces more speed-up on the ridgetops and on the lateral sides of the butte, compared to the linear upwind scheme (Figure 13d–f versus a–c).

All combinations of numerical settings produce more streaks throughout the flatter parts of the domain at Big Southern Butte than at Askervein Hill or Bolund Hill due to the presence of smaller topographic features surrounding the butte. High- and low-velocity streaks are visible upwind and to the sides of the butte, and are most prominent in the myKELU simulation (Figure 13a).

#### 5.3.2. Comparisons with Observations

For Big Southern Butte, we compare both wind speed and wind direction to observations along two transects, TSW and TWSW (Figures 14–16). The locations of the two transects are shown in Figure 14a. The profiles are not as smooth as at Askervein Hill or Bolund Hill because here the transects traverse multiple ridges and valleys on the butte. Figure 14b–c shows the terrain profiles along the two transects. Transect TSW has a steep approach to a ridge line, then traverses some small terrain features without substantial net elevation change, then has another steep approach to the highest point on the transect, followed by a steep descent down the northeast side of the butte (Figure 14b). Transect TWSW has a steeper and smoother approach to the highest point on the transect, followed by a steep

descent, which traverses one substantial valley about halfway down the butte (Figure 14c). Terrain representation in the CFD mesh is addressed in Section 5.6.

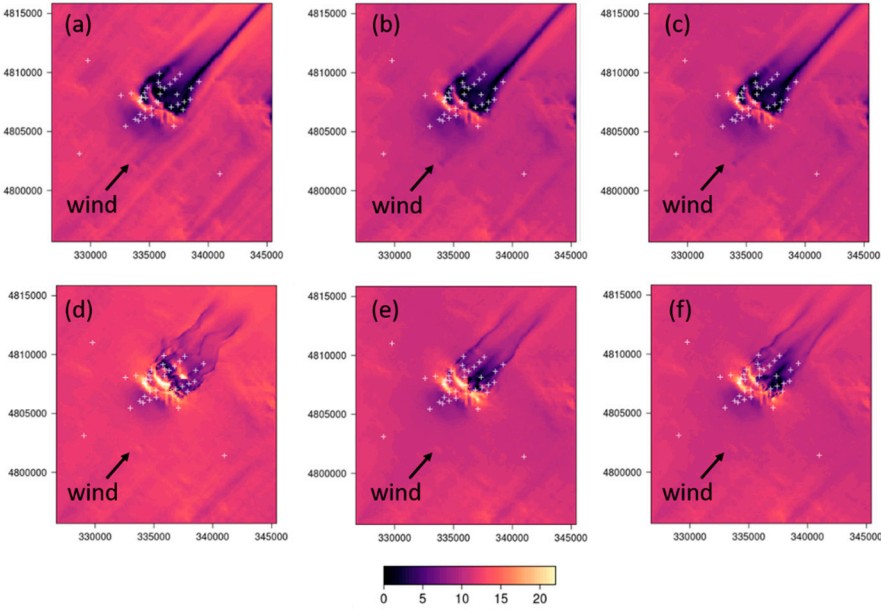

**Figure 13.** CFD-predicted wind speeds in m s$^{-1}$ at 3 m above ground level over Big Southern Butte using (**a**) myKELU; (**b**) KELU; (**c**) RNGKELU; (**d**) myKEQUICK; (**e**) KEQUICK; (**f**) RNGKEQUICK. White crosses indicate measurement locations. Black arrows denote the prevailing wind direction. Axes labels are in meters.

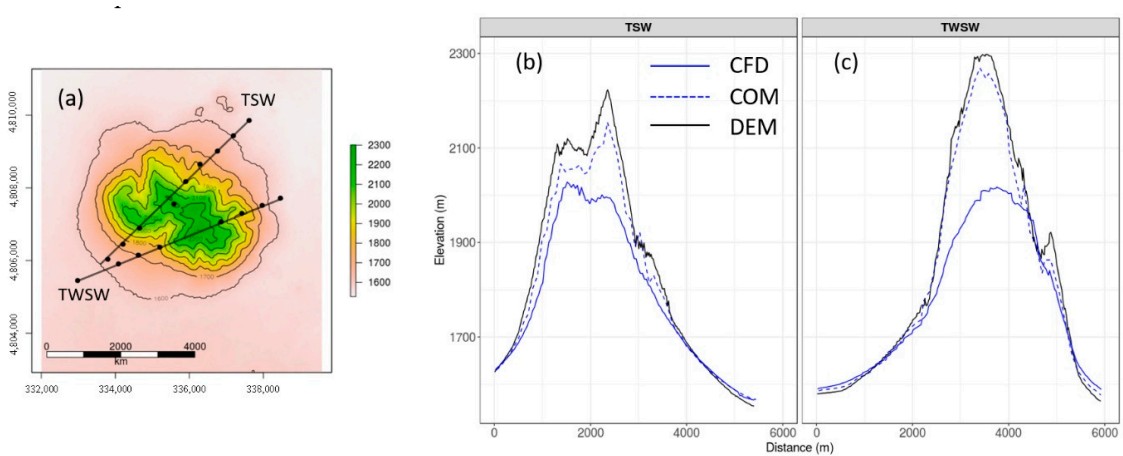

**Figure 14.** (**a**) Location of the TSW and TWSW transects and terrain representation in the meshes used for the CFD and COM simulations, along the (**b**) TSW and (**c**) TWSW transect. DEM is the digital elevation model.

The linear upwind simulations compare better with the observed speed-up than the QUICK simulations on the TSW transect (Figure 15a) and on the lee side of the TWSW transect (Figure 15b). The linear upwind simulations under-predict speed-up on the windward side of TWSW (Figure 15b). The QUICK simulations over-predict at the ridgetop locations and for most locations on the lee side of the transects. The COM solver predicts a smaller range of speed-up along both transects compared to the CFD simulations. The COM solver under-predicts on the windward side and over-predicts on the lee side of both transects (Figure 15).

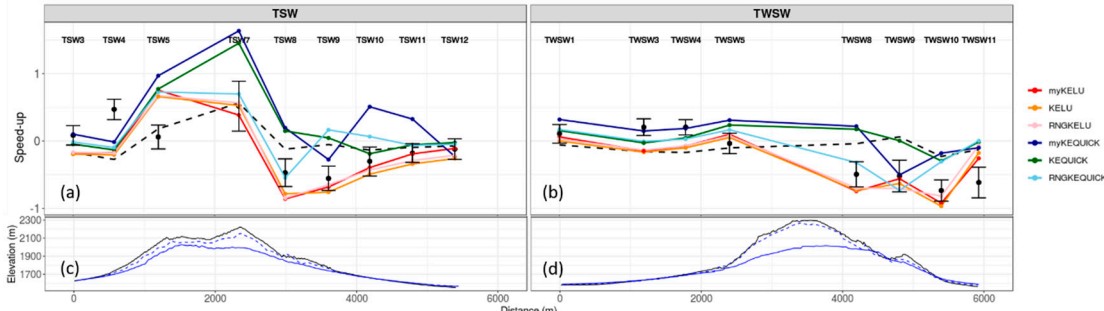

**Figure 15.** Model comparisons with observed speed-up at Big Southern Butte along transect (**a**) TSW and (**b**) TWSW. DEM and terrain representation in the meshes along transect (**c**) TSW and (**d**) TWSW as shown in Figure 15b,c. The x-axis is distance along the transect. The y-axis in (**a**) and (**b**) is the fractional speed-up, relative to the observed speed at a reference station upwind. Black circles are observed data. Error bars indicate plus and minus one standard deviation. The black dashed lines are the COM solver results.

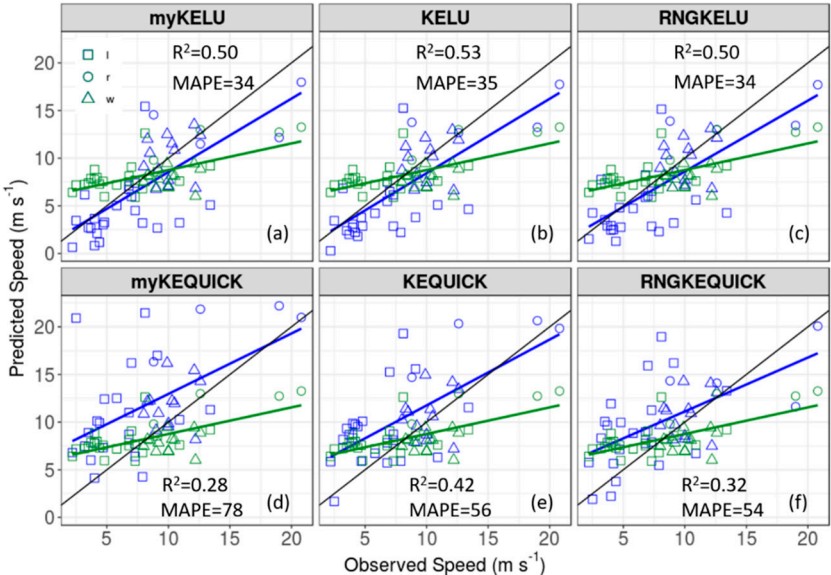

**Figure 16.** Observed versus predicted wind speeds at Big Southern Butte using (**a**) myKELU; (**b**) KELU; (**c**) RNGKELU; (**d**) myKEQUICK; (**e**) KEQUICK; (**f**) RNGKEQUICK. Blue symbols are for the CFD solver and green symbols are for the COM solver. The blue and green lines represent the ordinary least squares line of best fit for the CFD and COM solver, respectively. The black line is the 1:1 line. The mean absolute percent error (MAPE) and coefficient of determination ($R^2$) for the COM solver are 46 and 0.39, respectively.

The simulations using the linear upwind scheme have the lowest RMSE, MBE, and MAPE in wind speed of the CFD simulations at Big Southern Butte (Table 7; Figure 16). The myKELU, KELU, and RNGKELU, all have similar, and lower, MAPEs (34%, 35%, and 34%, respectively) than the myKEQUICK, KEQUICK, and RNGKEQUICK (78%, 56%, and 54%, respectively) and COM (46%) simulations (Figure 16). Inspection of the observed versus predicted regression lines shows that the linear upwind simulations also more closely approximate the 1:1 line. The COM solver over-predicts at the lower speeds and under-predicts at the higher speeds, with a regression line that bisects the 1:1 line nearly in the middle with a fairly flat slope. The linear upwind scheme predicts the lower speeds well and slightly under-predicts at higher speeds (Figure 16a–c). The QUICK scheme over-predicts at the lower speeds, which is consistent with results presented earlier, which showed that QUICK over-predicts on the lee side of the butte and under-predicts at only the highest speeds (Figure 16d–f). The KELU scheme has the closest approximation to the 1:1 line, the best regression fit ($R^2 = 0.53$), and

the lowest MAPE (35%, essentially the same as that for the myKELU and RNGKELU schemes) and can be considered the best model for this site.

**Table 7.** Model root mean square error (RMSE), mean bias error (MBE), and mean absolute percent error (MAPE) for wind speeds at windward (w), ridgetop (r), and leeward (l) sensor locations at Big Southern Butte. Positive MBE indicates model over-prediction. Bold font indicates the lowest error at each location.

| Location | Settings [1] | RMSE | MBE | MAPE |
|---|---|---|---|---|
| w | LU | **2.35** | **−0.30** | **19** |
|   | QUICK | 2.65 | 0.98 | 22 |
|   | COM | 2.70 | −2.17 | 20 |
| r | LU | **4.31** | **−1.00** | 28 |
|   | QUICK | 5.31 | 2.78 | 36 |
|   | COM | 4.93 | −3.11 | **21** |
| l | LU | 3.66 | **−1.55** | **44** |
|   | QUICK | 5.50 | 3.48 | 92 |
|   | COM | **3.16** | 1.82 | 65 |

[1] LU is linear upwind; QUICK is the Quadratic Upstream Interpolation for Convective Kinematics, and COM is conservation of mass.

The error bars for wind direction are notably larger on the lee side of the transects than on the windward side (Figure 17). The observed lee side flow is highly unsteady, with 180° fluctuations in wind direction at some locations over the 10 min averaging period (Figure 17). These fluctuations in wind direction correspond to enhanced turbulence associated with a lee side wake zone [5,22]. The observed mean southwest wind direction and smaller error bars at the last two locations on transect TSW, TSW11 and TSW12, suggest these locations are located outside of the wake zone (Figure 17a). Observed wind speeds are also higher at TSW11 and TSW12 than at the other lee side locations, closer to the butte (Figure 15a), further suggesting these locations are outside of the wake zone. In contrast, transect TWSW does not appear to extend beyond the wake zone (Figures 15b and 17b).

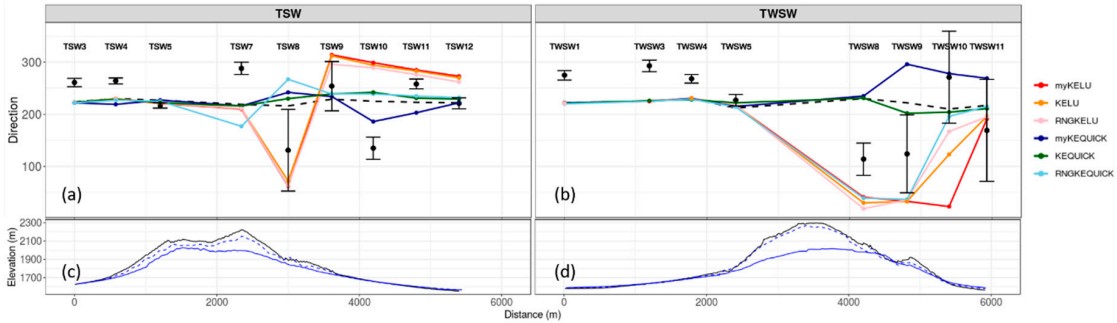

**Figure 17.** Model comparisons to observed wind directions at Big Southern Butte along transect (**a**) TSW and (**b**) TWSW. DEM and terrain representation in the meshes along transect (**c**) TSW and (**d**) TWSW, as shown in Figure 15b,c. The x-axis in is distance along the transect. Black circles are observed data. Error bars indicate plus and minus one standard deviation. The black dashed lines are the COM solver results.

The linear upwind scheme produces a larger range of wind directions along the two transects than the QUICK scheme does (Figure 17). This is consistent with the results previously discussed, which show that the linear upwind scheme produces larger and more coherent lee side regions of reduced velocities. The QUICK scheme, in contrast, produces narrower, shorter (in the streamwise direction) regions of reduced velocities (Figure 13). The COM solver simulates little change in wind direction over the two transects (Figure 17).

### 5.4. Summary Across Field Sites

Combining the data from all three field sites confirms that the choice of discretization scheme has a larger effect on wind speed error than the choice of turbulence model (relative importance of 20% versus 12%). The biggest difference in wind speed error between the discretization schemes is at the lee side locations where, on average, the QUICK scheme over-predicts by 3.0 m s$^{-1}$ and the linear upwind scheme under-predicts by 2.1 m s$^{-1}$ ($p < 0.0001$). The effect of the turbulence model on wind speed error was only significant when using the QUICK scheme, where the myKE model had the highest over-prediction of 1.9 m s$^{-1}$, compared to the KE model over-prediction of 0.78 m s$^{-1}$ ($p = 0.0037$), and the RNGKE over-prediction of 0.59 m s$^{-1}$ ($p = 0.001$), when averaged over all locations.

Although the results from the three field sites were mixed in terms of identifying the best combination of CFD settings, there is evidence to suggest that the linear upwind scheme may produce the best results when viewed over the entire range of data (Figure 18; Table 8). When data from all three sites are combined, and the three turbulence models are pooled together, the linear upwind scheme has the lowest MAPE of 27%, versus 35% for QUICK (Table 8), and the best ordinary least squares line fit ($R^2 = 0.63$ versus 0.46, Figure 18).

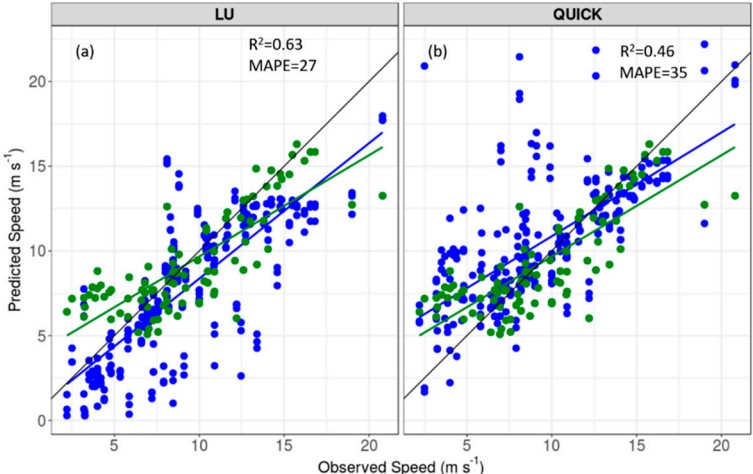

**Figure 18.** Observed versus predicted wind speeds at all sites using the (**a**) linear upwind and (**b**) QUICK discretization schemes. Blue symbols are for the CFD solver and green symbols are for the COM solver. The blue and green lines represent the ordinary least squares line of best fit for the CFD and COM solver, respectively. The black line is the 1:1 line. The mean absolute percent error (MAPE) and coefficient of determination ($R^2$) for the COM solver are 29 and 0.60, respectively.

The differences between the COM and the CFD solver are most apparent at the windward and leeward locations (Table 9). When averaged across all CFD settings, both the COM and CFD solvers over-predict wind speed on the lee side of the hill and under-predict on the windward side, with the CFD solver having significantly lower errors at both locations (lee: 1.72 versus 0.43 m s$^{-1}$, $p < 0.0001$; windward: −1.75 versus −0.18 m s$^{-1}$, $p < 0.0001$). In both cases, the CFD solver produced a smaller MAPE as compared to the COM solver (Table 9). However, at the ridgetop locations, the two solvers produced similar errors in wind speed, with the COM solver having the lowest MAPE at 12%. These results suggest that the additional computational expense required for the CFD solver is warranted if lee side or windward predictions are of interest. In contrast, if ridgetop speed predictions are the sole interest, the COM solver may be sufficient, as it produces statistically comparable predictions at ridgetop locations.

**Table 8.** Model root mean square error (RMSE), mean bias error (MBE), and mean absolute percent error (MAPE) for wind speeds at all locations at all sites. Positive MBE indicates model over-prediction. Bold font indicates the lowest error at each location.

| Settings | RMSE | MBE | MAPE |
|---|---|---|---|
| LU | 3.0 | −1.5 | **27** |
| QUICK | 3.3 | 1.1 | 35 |
| COM | **2.4** | **−0.11** | 29 |

LU is linear upwind; QUICK is the Quadratic Upstream Interpolation for Convective Kinematics, and COM is conservation of mass.

**Table 9.** Model root mean square error (RMSE), mean bias error (MBE), and mean absolute percent error (MAPE) for wind speeds at windward (w), ridgetop (r), and leeward (l) sensor locations for all sites. Positive MBE indicates model over-prediction. Bold font indicates the lowest error at each location.

| Location | Settings | RMSE | MBE | MAPE |
|---|---|---|---|---|
| w | LU | **1.75** | −0.65 | 14 |
|   | QUICK | 1.81 | **0.29** | **13** |
|   | COM | 2.21 | −1.75 | 18 |
| r | LU | 2.90 | −1.80 | 19 |
|   | QUICK | 2.32 | **−0.19** | 13 |
|   | COM | **2.18** | −0.38 | **12** |
| l | LU | 3.86 | −2.12 | **46** |
|   | QUICK | 4.76 | 2.99 | 76 |
|   | COM | **2.84** | **1.72** | 55 |

LU is linear upwind; QUICK is the Quadratic Upstream Interpolation for Convective Kinematics, and COM is conservation of mass.

## 5.5. Computational Expense Considerations

We have shown that the CFD solver produces significantly lower error in wind speed predictions on the windward and lee side locations, compared to the COM solver. We also compared against previously published LES results at two of the field sites and found that, although LES compared better with measurements in some cases, the CFD predictions generally fell within the bounds of the LES wind speed predictions. Whether these differences are large enough to be of practical importance to a user is a separate question, and more difficult to answer. The answer likely depends on several factors, including the intended use of the simulations, how precisely the input data are known, the computational resources available, and whether there are temporal constraints.

In wildland fire applications there is often considerable uncertainty in the input data, limited computational resources, and a need for predictions in very short timeframes (e.g., minutes to hours). Table 10 shows the computational requirements for the COM, CFD, and LES solutions. The COM solver is the fastest, with simulation times averaging about 10 s on a typical personal computer. The CFD solver is the next computationally efficient solver, with simulation times averaging about 5.5 min on a typical personal computer. Both would generally be acceptable timeframes in wildland fire incidents, depending on the modeling objectives (one exception might be if many simulations were needed for statistical analysis). The reported LES simulation time for Bolund Hill was 40 days using 512 processors, which is nearly 8000 times slower than the average CFD simulation time using 128 times the computing power; these computational demands are well beyond what operational fire managers have access to for their work.

Another crucial factor is user training. Fire managers do not typically have formal training in meteorology, engineering, or computer science. The models and tools that they use cannot require expertise in specialized fields and must be simple enough to be taught in the standardized training format used by wildland fire management. WindNinja is specifically designed to internally handle (without user interaction) the needed data assimilation, pre-processing, meshing, initialization, and

post-processing for the user. A typical fire manager would not have the expertise, let alone the needed computational resources or time, to run LES.

**Table 10.** Computational expense required for the COM, CFD, and LES simulations.

| Solver [1] | Simulation Time (min) | Number of Cells in Mesh | Number of Processors |
|---|---|---|---|
| | | Askervein Hill | |
| COM | 0.17 | 20K | 4 |
| CFD | 4.2 | 100K | 4 |
| LES [2] | - | - | - |
| | | Bolund Hill | |
| COM | 0.17 | 20K | 4 |
| CFD | 7.3 | 100K | 4 |
| LES [3] | 57,600 | 2.9M | 512 |
| | | Big Southern Butte | |
| COM | 0.16 | 20K | 4 |
| CFD | 4.9 | 100K | 4 |
| LES [4] | - | - | - |

[1] COM and CFD simulations run on a Thinkmate desktop with 3.47GHz Intel Xeon X5677 CPUs. [2] Askervein Hill LES simulation times were not reported in Golaz et al. [36]. [3] Bolund Hill LES simulation times reported in Vuorinen et al. [37]. [4] Unable to find published LES results for Big Southern Butte.

Ultimately, users should consider the tradeoff between accuracy and computational demand for their application. For wildland fire managers, we recommend using the WindNinja CFD solver whenever possible. One exception might be if only ridge-top speed-up is of interest to the user; in this case, the COM solver should give similar results and would be an acceptable choice.

*5.6. The Computational Mesh and Terrain Representation*

The current CFD meshing procedure is robust and has many desirable characteristics, including near-ground cells aligned with the terrain and smaller cells near the ground, where gradients are largest, but it also has several deficiencies. These deficiencies include that the height of the near-ground cell is dependent on the size of the domain, and that the transition between the coarse and fine cells (which is often near the ground) is bridged by irregular, wedge-shaped cells that are not terrain-following, and the cell height is forced to equal the horizontal cell size near the ground, which results in fine horizontal resolution, but relatively coarse vertical resolution.

The effects of the wedge-shaped cells can be seen in the oscillating speed-up lines, where we sample some of these cells in the Bolund Hill mesh (Figure 13b). This happens when sampling is done through the transition region between the coarse and fine cells; the cell centers of the wedge-shaped cells are not necessarily in the same plane, and field interpolation through that plane can lead to oscillations in the sampled field. Unfortunately, as configured, our meshing procedure does not allow us to specify the location of this transition region; the location is governed by the size of the domain and the number of cells allocated for the mesh.

Another limitation in the current meshing procedure is related to the use of moveDynamicMesh to stretch the lower part of the mesh down to the terrain. Mesh movement is done before mesh refinement, primarily for speed (mesh motion can be faster with larger cells). This can introduce potentially large errors in terrain representation, since relatively coarse cells are used to approximate the underlying terrain. At Big Southern Butte, these errors in terrain representation are large compared to terrain representation in the COM mesh (Figure 16a,b). This is also likely to explain why we did not observe appreciable improvement in results when the mesh count was increased beyond 100k cells. We suspect that errors related to terrain representation in the CFD mesh may be one of the largest sources of error in the model, particularly when the terrain is highly complex.

We have investigated many combinations of OpenFOAM meshing utilities, including snappyHexMesh and various methods of applying refineMesh, but have not found an alternate meshing method, that is both robust and superior in terms of terrain representation and mesh quality, than what is currently implemented. Other options include writing custom mesh generation code or using third-party mesh generation software. Future work will explore these alternative meshing options.

## 6. Conclusions

A new CFD solver recently implemented in the WindNinja wind modeling framework has been described. Results from the CFD solver are compared against observations from three field campaigns, the COM solver in WindNinja, and previous LES simulations. Six combinations of numerical settings were investigated. The sensitivity to these settings is likely a function of the mesh resolution near the ground. The meshes investigated in this study are representative of what would typically be used in operational wildland fire applications. The meshes are relatively coarse, in order to meet operational emergency response constraints. We anticipate that the sensitivity to the discretization scheme and turbulence model would change if finer resolution could be obtained in the near-ground layers of the mesh, either from an increase in the total cell count or strategic placement of additional cells in the near-ground region and reduction of cells elsewhere in the mesh. Noting that our results are a function of the meshes used in this work, which are representative of those that would be used by typical WindNinja users, the main findings of this work are:

1. Overall, the CFD solver performs better than the COM solver at all sites investigated, particularly at the windward and lee side locations. For ridgetop locations, however, the COM solver produces statistically comparable wind speed predictions and, thus, if ridgetop predictions are solely of interest, the additional computational expense required for the CFD solver may not be necessary.
2. The choice of discretization scheme used for the advection term in the momentum equation has a bigger effect on wind speed error than the choice of turbulence model.
3. The linear upwind scheme (and the QUICK scheme to a lesser degree) produces low-velocity streaks in the flow field that extend far downwind of terrain obstacles. The streaks are associated with the convergence of paired counter-rotating vortices in the wake zone, induced by the terrain. Future work should further investigate the initiation, dynamics, and structure of these paired vortices and associated streaks in the mean flow to assess their representation in time-averaged numerical models.
4. The QUICK scheme produces higher speed-up over terrain features, higher lee side velocities, and less lee side variability in wind direction, as compared to the linear upwind scheme.
5. Results are mixed among the locations and cases examined at each site, but the linear upwind discretization scheme performs better than the QUICK scheme overall, in terms of the MAPE.
6. Sensitivity to the turbulence model choice is small compared to the choice of discretization scheme, so the choice of turbulence model is less important than choice of discretization scheme. The three turbulence models had nearly identical MAPE at Big Southern Butte when the linear upwind scheme was used. Without definitive quantitative results, other criteria must be used to select a turbulence model. We suspect that the most well-defined, low-velocity streaks produced by the myKE simulations may be an artifact of insufficient turbulent diffusion in the model. The standard KE model produced less well-defined streaks and is a slightly simpler formulation than the RNGKE model. Based on this, we recommend the KELU combination be used in WindNinja until further data is available to significantly identify differences among the turbulence models.
7. LES simulations visually compare better with the observations at Askervein Hill, particularly on the lee side, but CFD solver results fell within the bounds of previously reported LES results at Bolund Hill. Model users and developers should carefully consider whether potentially modest gains in mean wind speed predictions warrant the substantial increase in computational cost and complexity of LES. This recommendation especially applies to emergency response-type

situations, such as wildland fire, where time frames are short and uncertainty related to input conditions (initial wind, vegetation structure, etc.) is high.

8.  The current meshing procedure results in undesirable wedge-type cells at the interface between the coarse and the refined mesh at the surface and occasionally in the near-ground layer. The meshing procedure can be improved to better represent the terrain. Ideally, the mesh would (1) be terrain-following near the surface with horizontal grid lines gradually becoming normal to the z-axis at the top of the domain and hexahedral cells throughout; (2) have vertical grid lines that are perpendicular to the terrain near the ground, but gradually curve to become aligned with the z-axis (normal to the x–y plane) at the top of the domain; (3) have near-ground cells with much smaller cell heights than horizontal size to allow more vertical resolution at the surface without substantially increasing the total cell count.

These findings are important for WindNinja users as well as developers and users of other flow models designed to simulate atmospheric boundary layer winds over complex terrain. Future work will focus on improving the CFD meshing procedure, incorporation of non-neutral stability effects in the CFD solver, and continued evaluations over various types of complex terrain.

**Author Contributions:** Conceptualization, N.S.W. and J.M.F.; methodology, N.S.W. and J.M.F.; software, N.S.W. and J.M.F.; validation, N.S.W.; formal analysis, N.S.W. and W.G.P.; investigation, N.S.W.; resources, B.W.B.; data curation, N.S.W.; writing—original draft preparation, N.S.W.; writing—review and editing, J.M.F., W.G.P. and B.W.B.; visualization, N.S.W.; supervision, N.S.W. and J.M.F.; project administration, N.S.W. and J.M.F.; funding acquisition, B.W.B. and N.S.W.

**Funding:** This research was funded by the United States Department of Agriculture Forest Service office of the Deputy Chief for Research and the Rocky Mountain Research Station.

**Conflicts of Interest:** The authors declare no conflict of interest.

## Appendix A

A grid dependence study was conducted for Case 3 at Bolund Hill. We compare results from the mesh used in the current study (generated with the fine mesh setting in WindNinja, which uses 100K cells in the mesh) with those from meshes containing 2M and 4M cells. The mesh with 4M cells has a near-ground cell height 0.59 m. The major flow characteristics do not change when the cell count is increased from 100K to 4M; simulations with all three meshes produce reduce speeds in the approach flow to the hill, a speed up at the ridge top, and reduced speeds in the lee of the hill (Figures A1 and A2). Results from the QUICK scheme are more sensitive to the mesh resolution than those from the LU scheme (Figures A1 and A2). This suggest that the LU scheme may produce better results with coarse meshes compared to the QUICK scheme.

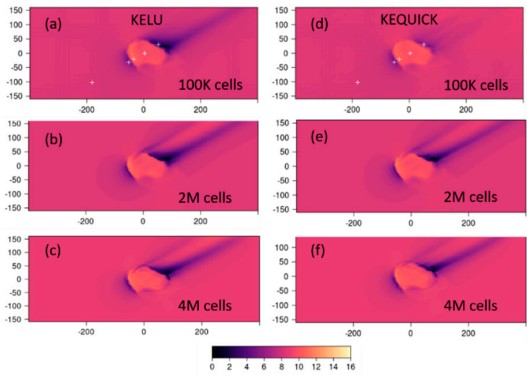

**Figure A1.** CFD-predicted wind speeds in m s$^{-1}$ at 5 m above ground level over Bolund Hill for Case 3, using (**a**) KELU with 100K cells; (**b**) KELU with 2M cells; (**c**) KELU with 4M cells; (**d**) KEQUICK with 100K cells; (**e**) KEQUICK with 2M cells; (**f**) KEQUICK with 4M cells. White crosses in the top panels indicate measurement locations. Axes labels are in meters.

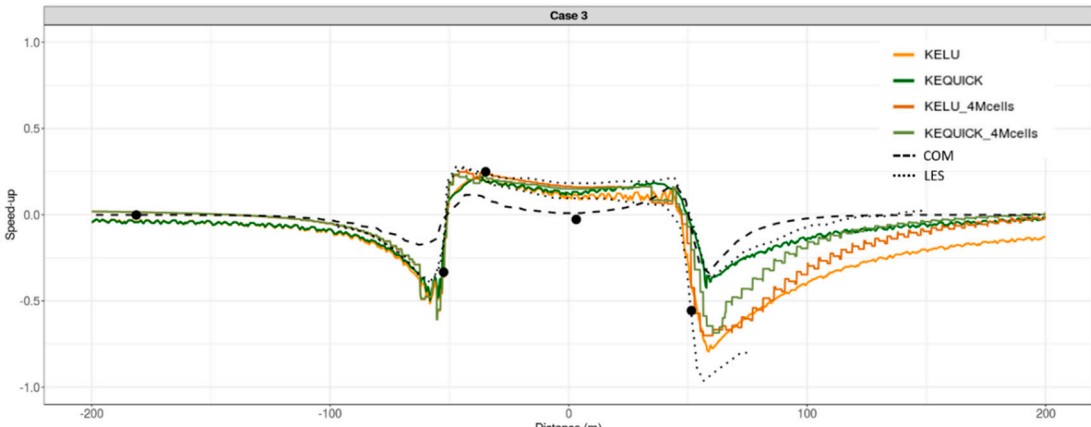

**Figure A2.** Model comparisons with observed data at 5 m above ground level at Bolund Hill for Case 3. KELU and KEQUICK are the results from the mesh used in this study with 100K cells; KELU_4Mcells and KEQUICK_4Mcells are the results from meshes with 4M cells. Black circles are observed data. The x-axis is distance along the transect. The y-axis is fractional speed-upc relative to the observed speed at a reference station upwind. The black dashed line is the COM solver results. Dotted black lines are LES results redrawn from Bechmann et al. [21] and Vuorinen et al. [37].

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
