# Peer review of "Development and Evaluation of a Reynolds-Averaged Navier–Stokes Solver in WindNinja for Operational Wildland Fire Applications"

_atmosphere, doi:10.3390/atmos10110672_

Round 1

Reviewer 1 Report

The present work deals with the application of a Navier-Stokes solver (OpenFOAM) to the calculation of the wind field over complex terrain and comparison with a mass-consistent solver.
The main deficiency of this work relies on the limitations of the mesh generation procedure, leading to equal mesh spacing along the 3 directions for the mesh volumes adjacent to the ground. This is totally inadequate to the present applications, since the attained mesh resolution in the vertical direction is far too coarse. The results presented by the authors confirm this. The Askervein and Bolund cases are well known by the reviewer. The strikes shown in the wind field visualization (figure 7 and figure 9) are much suspicious and not sufficiently explained. Those strikes do not show in high resolution simulations published by other authors (and also made by the reviewer) and could, thus, be due to discretization errors. Quantitative results provided by graphics in figure 8 (Askervein) show a very high discrepancy between measurements and simulations. High discrepancy is also evident in the Bolund simulations (figure 12). The reasons for this are:
Askervein: The horizontal spacing should be adequate (20 m) but the ground adjacent CV heigh (20 m) is far too large. CV height should be below 1 m.
Bolund: The horizontal spacing (3.8 m) is too large and the ground adjacent CV height (3.8 m) should be below 0.2 m.
Proper numerical simulations should include a mesh dependency study. Without a proper mesh, no valid conclusions can be taken. This invalidates, in my opinion, all conclusions regarding influence of turbulence model and influence of discretization scheme, that are presented in this work.
In summary, the results presented in this work are totally unreliable due to the lack of mesh independency. Therefore, I cannot recommend this work for publication.

Reviewer 2 Report

This is a well-written manuscript that details a performance evaluation of WindNinja. I learned quite a lot about the product while reviewing this manuscript and as a result I would feel more confident discussing the strengths and weaknesses of this product with users. I therefore feel this manuscript would be important to the fire science and user community and recommend that it be accepted for publication following the minor revisions indicated below.

Overall comments:

1) Some run-on sentences and questionable grammar choices appear at places throughout this manuscript. Examples have been noted below. While few if any of these sentences include incorrect information, the manner in which they are worded is potentially confusing, particularly for non-native English speakers. I feel these sentences should be rewritten for improved overall clarity.

2) I question the order in which the conclusions are presented, both in the manuscript and in the abstract. I find the result that the CFD solver is superior to the COM solver at upwind and downwind locations to be the most significant finding in this study. That it is “buried” at number 6 out of 8 bullets in the conclusion feels strange to me, as does its appearance following the discretization scheme in the abstract. If the authors disagree with my impression I can accept that, but my sense is that most readers would would find the improvement from the solver to be the most significant result here.

Specific comments:

Line 12: run-on sentence.

Line 16: run-on sentence

Line 18: Why are these settings more or less important than others in the solver? I don’t think “important” is necessary here.

Line 20: It is not clear, based on the abstract, where these findings came from. Are the choice of advection scheme and turbulence scheme the “two important settings”? If so, the authors should state this explicitly.

Lines 47-50: These sound more like sales pitches than scientific observations. What makes a framework “robust”? What is a “modern” GUI? How is “easy-to-use” determined? None of these superlatives are necessary to make the point the authors are making, and I would argue they should be eliminated.

Line 53: Similarly, what does “embedded directly” mean? Is it possible to embed code indirectly? Less sales speak and more factual wording would be desirable here.

Line 55: run-on sentence

Line 59-61: poor grammar

Line 64: run on sentence

Line 70: questionable/confusing grammar (a list followed by a phrase that only applies to the last entry in the list)

Line 75: run-on sentence

Line 76: grammar – items in the list are not parallel

Line 80: grammar – items in the list are not parallel

Line 82: grammar

Line 87: run-on sentence

Line 90: strange punctuation

Line 182: I believe a comma is needed after ‘fine’ in this sentence.

Lines 221-225: Grammar, in both sentences, could be clearer.

Line 279: run-on sentence

Line 475: There should be a unit in the y-axis of Figure 8 (m s-1).

Line 541: run-on sentence

Line 544: The observations (black circles) are rather difficult to see in Fig. 12.

Line 548: “reasonable job” is subjective. I would just say they all reproduce the reduced speed in the approach flow.

Line 619: DEM needs to be defined in Fig. 14. Also, it seems strange to me that there is no corresponding figure indicating differences in terrain representation for the previous cases. Is there a reason why it is important here but was not important for the other cases? If so, this reason should be stated.

Line 632: The labels in Fig. 15 (and similarly in Fig. 17) are too small. It is too difficult to read the legends and determine which line corresponds to which model.

Line 735: “fell within the ballpark” is both subjective and colloquial. The authors should use the wording that appears in line 834 in the conclusions here as well.

Line 776: “high” should be “fine” (to contrast with “coarse” later in the sentence).

Line 792-793: This is an awkward sentence construction. I would recommend that elements of the sentence be reordered so it is more clear to the reader (i.e. move the phrase separated by commas to the end of the sentence).

Line 801-804: run on sentence (this one should just be a list of the three comparisons without the “as well as”, which is unnecessary and confusing).

Line 807: “at least”? It is not clear to me that this is necessary. Also, “is true” is more chatty than it is precise scientific language. There is no truth here (although there might be beauty :) ).

Line 837: more “truth” here. I recommend an alternative word choice.

Reviewer 3 Report

I recommend this paper for publication. This is one of the best papers I have reviewed so far. 

I have one question, but it is more to satisfy my curiosity, rather than a comment on the paper. How does the CFD model deal with forests/canopy? Is it just through roughness, or do you modify the boundary layer sub-model?

Round 2

Reviewer 1 Report

After my first review, the authors improved the manuscript by leaving clear that present simulations are much limited through mesh resolution. I think that now the reader will be aware of that. Furthermore, comparisons with Bolund and Askervein data leave clear that the quality of the results is rather low, although with some value for operational purposes. Nevertheless, I would like to stress that not being able to control in an independent way the vertical and horizontal resolution is a big disadvantage of this approach, in particular when dealing with atmospheric flows.

Line 168, Table 1: CMU coefficient is usually taken as 0.033 for atmospheric flows. The authors should justify why they adopted the standard value 0.09.

Lines 214, 215 – Ground boundary conditions are referred through the OpenFOAM terminology. This gives little information to the reader that is not familiar with the software.

Line 215: The authors mention zero value as boundary condition for velocity (Dirichlet-type boundary condition). How do this cope with the roughness information at the ground? The usual assignment for velocity is a flux boundary condition through the shear stress, where ground roughness appears in the logarithmic law. Please clarify.

Figure A2 – There are 7 different results but the figure legend only shows 4 entries. Please specify the height above ground (it should be 5 m). The high frequency oscillating behavior of some results suggest some numerical problems. Please comment.
